# Dual-Channel Mid-Infrared Toroidal Metasurfaces for Wavefront Modulation and Imaging Applications

**DOI:** 10.3390/nano12193300

**Published:** 2022-09-22

**Authors:** Jingyu Zhang, Chang Liu, Hengli Feng, Dongchao Fang, Jincheng Wang, Zuoxin Zhang, Yachen Gao, Yang Gao

**Affiliations:** 1School of Electronic Engineering, Heilongjiang University, Harbin 150080, China; 2Heilongjiang Provincial Key Laboratory of Metamaterials Physics and Device, Heilongjiang University, Harbin 150080, China

**Keywords:** metasurface, two channels, programmable imaging, thermal imaging

## Abstract

In this paper, we propose a dual-channel mid-infrared toroidal metasurface that consists of split equilateral triangular rings. The electromagnetic responses are analyzed by the finite-difference-time-domain (FDTD) method and temporal coupled-mode theory (TCMT). The results show that one channel of the metasurface is insensitive to the polarization angle of the incident light and temperature, while the other channel is sensitive. The reflectance and resonance wavelength can be manipulated by the polarization angle and temperature independently. Based on such a mechanism, we propose metasurfaces for two-bit programmable imaging and thermal imaging. The metasurfaces are believed to have potential applications in information processing and thermal radiation manipulation.

## 1. Introduction

Composed of subwavelength antennas, metasurfaces are widely used to manipulate the polarization, phase, and amplitude of electromagnetic waves because metasurfaces have many advantages such as the small size and little loss [1]. Lots of devices based on metasurfaces have been proposed, e.g., absorbers [2], invisibility cloaks [3], special beam generators [4], programmable elements [5], and thermal imaging devices [6]. The design of programmable metasurfaces is an effective approach to suppressing the crosstalk [7,8,9,10,11], and thermal imaging metasurfaces can be used for temperature measurement [12], night-vision [13], and heat detection [14]. For example, Shang et al. designed a reconfigurable thermal metasurface to achieve arbitrary objects’ thermograms [15]. Alipour et al. proposed a bolometer sensor with high resistance and sensitivity [16]. Liu et al. illustrated a metal–liquid-crystal–metal metasurface to realize dynamic thermal camouflage [17]. With these advantages, metasurfaces are believed to have great potential applications.

Metasurfaces are mainly divided into sensitive and insensitive types. Sensitive metasurfaces can work as sensors [18,19], bolometers, and photodetectors. The tunability can be realized by introducing external excitation methods including electric [20], optical [21,22,23], mechanical controlling [24], and thermal tuning [25,26]. As for the insensitive type, they usually maintain a relatively stable performance to environmental changes. Recently, in order to meet various practical applications, metasurfaces with higher flexibility have become a research hotspot. Although both sensitive and insensitive metasurfaces have been proposed and extensively studied in recent years, metasurfaces with the ability to switch between sensitivity and insensitivity are rarely reported.

In this paper, a dual-channel mid-infrared toroidal metasurface that is composed of split equilateral triangular ring resonators is proposed. The reflectance spectrum of the metasurface is studied by both the finite-difference-time-domain (FDTD) method and temporal coupled-mode theory (TCMT). The results illustrate that the toroidal dipole metasurface has two channels. Channel A is insensitive to the polarization angle of the incident light and temperature, while channel B is sensitive to the polarization angle and temperature. Therefore, a two-bit programmable metasurface is achieved based on different sensitivities of the two channels to polarization angles. Furthermore, another metasurface is proposed for thermal imaging based on the different sensitivities of the two channels to temperature. The proposed metasurfaces offer alternative approaches to software-controlled digital metasurfaces and the photonic thermotropic.

## 2. Materials and Methods

The proposed metasurface is composed of periodical arraying unit cells and the unit cell is depicted in Figure 1a. A calcium fluoride (CaF_2_) substrate is fabricated by chemical mechanical polishing. Then, a split equilateral triangular silicon (Si) ring is etched on the substrate [27] and the gap of the ring is filled with GST [28]. In addition, the ring is immersed in air. The thicknesses of the split ring and the gap are D = D_1_ = 0.5 μm. There is an *X-Y* plane view of the unit cell in Figure 1b. The lattice constant of the unit cell is *K* = 3 μm. The outer side length of the triangular ring is *L* = 1.74 μm, whereas the inner side length is *L*_1_ = 0.87 μm. The length and width of the gap are *H* = 0.7 μm and *W* = 0.27 μm, respectively. In Figure 1c, a linearly polarized incident plane wave illuminates vertically on the unit cell and a thermal field is applied on the unit cell. In other words, the unit cell is heated by an external heat source. As is shown in Figure 1d, the incidence angle *θ* of the incident plane wave is defined in the *X-Z* plane. The incidence angle *θ* varies between 0° and 60° and the initial incidence angle *θ* is set as 0°. Optical responses of the metasurface are simulated by FDTD. Periodic boundary conditions are chosen in the *X-* and *Y*- directions and perfectly matched layers are set in the *Z-* direction, and this simulation model is semi-infinite in the *Z-* direction. The refractive indexes of Si and CaF_2_ are obtained from Ref. [29].

Affected by an external thermal field [30], GST is transformed from amorphous to crystalline. Permittivities of GST at amorphous ($\varepsilon_{aGST}$) and crystalline ($\varepsilon_{cGST}$) states are derived from [31]. The permittivity of GST ($\varepsilon_{GST}$) related with the crystalline fraction *m* (0 ≤ *m* ≤ 1) is described as [32]:(1)$$\frac{\varepsilon_{GST}(m)-1}{\varepsilon_{GST}(m)+2}=m\times \frac{\varepsilon_{cGST}(m)-1}{\varepsilon_{cGST}(m)+2}+(1-m)\times \frac{\varepsilon_{aGST}(m)-1}{\varepsilon_{aGST}(m)+2}$$

In Figure 2, the temperature is set as 100° and GST is in an amorphous state. The simulation results in Figure 2 are obtained when a *Y*-direction-linearly polarized incident plane wave illuminates vertically on the metasurface. The structural parameters in Figure 2 are the same as those in Figure 1. In Figure 2a, reflectance spectra of the metasurface are shown and the resonance wavelengths are at 3.51 μm and 3.82 μm, respectively. To further understand the resonances in the metasurface, TCMT is used to theoretically analyze the interactions between the metasurface and the incident light. In TCMT, normalized resonant amplitudes $a_{1}$ and $a_{2}$ as functions of the input amplitudes ($S_{1+}$ and $S_{2+}$) and the output amplitudes ($S_{1-}$ and $S_{2-}$) are expressed as [33,34]:(2)$$\frac{d}{dt}\left(\begin{array}{c} a_{1}\\ a_{2} \end{array}\right)=(j\Lambda -\gamma_{i}-\gamma_{e})\left(\begin{array}{c} a_{1}\\ a_{2} \end{array}\right)+B_{1}\left(\begin{array}{c} S_{1+}\\ S_{2+} \end{array}\right)$$
(3)$$\left(\begin{array}{c} S_{1-}\\ S_{2-} \end{array}\right)=B_{1}^{T}\left(\begin{array}{c} a_{1}\\ a_{2} \end{array}\right)+B_{2}\left(\begin{array}{c} S_{1+}\\ S_{2+} \end{array}\right)$$
where $\Lambda =\left(\begin{array}{cc} w_{1} & -\kappa \\ -\kappa & w_{2} \end{array}\right)$, $\gamma_{i}=\left(\begin{array}{cc} \gamma_{i1} & 0\\ 0 & \gamma_{i2} \end{array}\right)$, $\gamma_{e}=\left(\begin{array}{cc} \gamma_{e1} & 0\\ 0 & \gamma_{e2} \end{array}\right)$, $B_{1}=\left(\begin{array}{cc} j\sqrt{\gamma_{e1}} & j\sqrt{\gamma_{e2}}\\ j\sqrt{\gamma_{e1}} & j\sqrt{\gamma_{e2}} \end{array}\right)$, and $B_{2}=\left(\begin{array}{cc} 0 & 1\\ 1 & 0 \end{array}\right)$. $w_{1}$ and $w_{2}$ represent the resonance frequencies; $\gamma_{i1}$, $\gamma_{i2}$, $\gamma_{e1}$, and $\gamma_{e2}$ are the intrinsic decay rates and the external decay rates of different resonant modes, respectively. The coupling coefficient κ illustrates the coupling between two resonant modes. $B_{1}$ shows the coupling matrix between two resonant modes and the incident light. $B_{1}^{T}$ is the transpose matrix of $B_{1}$. $B_{2}$ is the direct scattering matrix. When the relationships between $B_{1}$, $B_{2}$, and $\gamma_{e}$ are satisfied as $B_{2}B_{1}^{*}=-B_{1}$ and $B_{1}^{*}B_{1}=2\gamma_{e}$, combined with Equations (2)–(4), the reflectance *R* can be concluded as:(4)$$R=1-T=1-\frac{(jw-jw_{1}+\gamma_{i1})(jw-jw_{2}+\gamma_{i2})+\kappa^{2}}{(jw-jw_{1}+\gamma_{i1}+\gamma_{e1})(jw-jw_{2}+\gamma_{i2}+\gamma_{e2})+\kappa^{2}}$$

In Equation (4), because Si and CaF_2_ have quite small optical loss [29], the absorption is small enough to ignore compared with reflectance. The calculation results are depicted with the black dotted line in Figure 2a, which is in good agreement with the simulation results. 

In Figure 2b, Cartesian multipole moments are shown to measure the contributions to the optical resonances of the metasurface. The electric dipole P; magnetic dipole M; toroidal dipole T; electric quadrupole $Q_{e}$; magnetic quadrupole $Q_{m}$; scattering powers of each multipole moment $I_{P}$, $I_{M}$, $I_{T}$, $I_{Q_{e}}$, $I_{Q_{m}}$; reflectance R are obtained as follows [35,36,37,38,39]:(5)$$J(t)=iw\varepsilon_{0}(\varepsilon_{d}-1)E(t)$$
(6)$$P=\frac{1}{iw}\int J(t)\;d^{3}t,\;I_{P}=\frac{2w^{4}}{3c^{3}}{\left|P\right|}^{2}$$
(7)$$M=\frac{1}{2c}\int \left(t\times J(t)\right)\;d^{3}t,\;I_{M}=\frac{2w^{4}}{3c^{3}}{\left|M\right|}^{2}$$
(8)$$T=\frac{1}{10c}\int \left[(t\cdot J(t))t-2t^{2}J(t)\right]\;d^{3}t,\;I_{T}=\frac{2w^{6}}{3c^{5}}{\left|T\right|}^{2}$$
(9)$$Q_{e}=\frac{1}{2iw}\int [t_{\alpha }J_{\beta }+t_{\beta }J_{\alpha }-\frac{2}{3}(t\cdot J(t))\varphi_{\alpha ,\beta }]\;d^{3}t,\;I_{Q_{e}}=\frac{w^{6}}{5c^{5}}\sum {\left|Q_{e}\right|}^{2}$$
(10)$$Q_{m}=\frac{1}{3c}\int \left[{\left(t\times J(t)\right)}_{\alpha }t_{\beta }+[{\left(t\times J(t)\right)}_{\beta }t_{\alpha }]\right]d^{3}t,\;I_{Q_{m}}=\frac{w^{6}}{40c^{5}}\sum {\left|Q_{m}\right|}^{2}$$
(11)$$R={\left|r\right|}^{2}={\left|\frac{\mu_{0}e^{ik_{0}}c^{2}}{2K^{2}E_{0}}\left[-ik_{0}P_{//}-ik_{0}\times M_{//}-k_{0}^{2}T_{//}-k_{0}{\left(k_{0}\cdot Q_{e}\right)}_{//}-\frac{k_{0}}{2}\times {\left(k_{0}\cdot Q_{m}\right)}_{//}\right]\right|}^{2}$$
where E(t) denotes the electric field of a metasurface unit cell extracted from FDTD simulation; $\varepsilon_{d}$ represents the relative permittivity of vacuum; c expresses the speed of light in vacuum; Cartesian coordinates α, β = x, y; $\mu_{0}$ is the magnetic permeability of vacuum; $k_{0}$ is the wave vector; $E_{0}$ is the electric field of the incident light; subscript (…)_//_ denotes the projection of the multipole moments into the *X-Y* plane; the direction of $k_{0}$ is the propagation of the incident light. Equation (11) illustrates the relationship between reflectance R, reflection coefficient r, and multipole moments. Here, we mainly study the contributions of P, M, T, $Q_{e}$, and $Q_{m}$, and higher-order multipoles are neglected. As shown in Figure 2b, two obvious toroidal dipole peaks are observed. The positions well-coincide with the reflection peaks in Figure 2a. As a result, the reflection peaks are mostly attributed to the toroidal dipoles. The far-field distributions at the resonance wavelengths are depicted in Figure 2c. The normalized electric field intensity distributions of the unit cell in the *X-Y* plane at the resonance wavelengths are shown in Figure 2d,e when *Z* = 0. The normalized electric field intensity distributions of the unit cell in the *Y-Z* plane at resonance wavelengths are shown in Figure 2f,g when *X* = 1.5 μm.

## 3. Results and Discussion

### 3.1. Wavefront Modulation

The impacts on reflectance spectra caused by the changes in geometrical parameters are shown in Figure 3. The split triangular ring can be seen as an L-C circuit of which the inductance and capacitance are inversely proportional to the resonance wavelengths of the metasurface [26]. Figure 3a shows *W* changing from 0.25 μm to 0.29 μm and the other structural parameters are the same as the parameters in. Because *W* represents the position occupied by the current flow, increasing *W* leads to a decrease in inductance of the gap of the ring and an obvious redshift in the right resonance wavelength. In Figure 3b, the two resonance wavelengths increase when *H* varies from 0.2 μm to 1.2 μm with the other parameters unchanged. An apparent redshift occurs in the right resonance wavelength with increasing reflectance. The gap can be considered as a capacitor with both sides accumulating negative and positive charges. As for a capacitor, an increment in the distance between the two electrodes leads to the growth of charging and discharging abilities. With the increase in *H*, the capacitance is reduced and the wavelength of the reflection peak is increased. Figure 3c illustrates that increasing *L* from 1.74 μm to 2.08 μm with the other parameters unchanged causes the reduction in the inductance of the ring and the redshifts in both resonance wavelengths with growing reflectance. As for Figure 3d, the growth of *L_1_* from 0.87 μm to 1.05 μm with the other parameters unchanged brings blueshifts in both resonance wavelengths without an apparent change in reflectance. This may be attributed to the growth of the induced current density of the ring.

Figure 4 shows a further study on the metasurface that has the same structural parameters as Figure 1. In Figure 4a, the reflectance spectra of the metasurface are evaluated at different polarization angles. The left peak is considered as channel A and the right peak is considered as channel B. When the polarization angle is 0°, it means that the electric field of a linearly polarized incident plane wave is parallel to the *Y*-direction, and when the polarization angle is 90°, the electric field is parallel to the *X-*direction. Increasing the polarization angle from 0° to 90°, from the *Y*-direction to the *X-*direction, the reflectance in channel A is invariant but the reflectance in channel B is reduced. Figure 4b shows the results calculated by TCMT, which has the same tendency with Figure 4a. Figure 4c shows normalized far-field scattering patterns of channel A with different polarization angles. Figure 4d shows the normalized far-field scattering patterns of channel B with different polarization angles. The reflected light is mainly limited in the ranges of 60° to 120° and 240° to 300° and the reflectance decreases with the change in polarization angle from 0° to 90°. With growth of the crystalline fraction of GST in the gap, the left resonance wavelength is fixed and the right has a redshift as shown in Figure 4e. Increasing the crystalline fraction causes growth of the effective permittivity of the gap. Growth of the effective permittivity can be reflected as the increase in *W* and *H* of the gap in Figure 3a,b. Therefore, changing the crystalline fraction has similar impacts on the reflectance spectrum of the metasurface as changing *W* and *H*. As is shown in Figure 4f, calculated by TCMT, the results have the same tendency as those in Figure 4e. In Figure 4g,h, the reflectance of the left and right resonances reduce when the crystalline fraction changes from 0 to 1. From Figure 4a to Figure 4h, it is concluded that channel A is insensitive to the change in polarization angle and crystalline fraction of GST. However, channel B is sensitive and the reflectance and resonance wavelength of channel B can be manipulated by the polarization angle and the crystalline fraction of GST independently. In Figure 4i, with the change in incidence angle *θ* from 0° to 60°, the two resonance wavelengths are fixed and the reflectance at the two resonance wavelengths is decreased. Figure 4j shows that the calculated results have the same tendency as those in Figure 4i. In Figure 4k,l, the reflectance of the left and right resonance wavelengths diminishes with the growth of incident angle *θ* from 0° to 60°. 

Because the reflection peaks are mainly attributed to toroidal dipoles in Figure 2b and toroidal dipoles are calculated by current density in Equation (8), the surface current distributions are introduced to illustrate the physical mechanism of Figure 4a. The surface current distributions of the unit cell with different channels and polarizations are shown in Figure 5. The surface current distributions are evaluated at *Z* = 0. The color scale represents the normalized current density. White arrows in Figure 5 represent the directions of the electric field of the unit cell. According to Equation (5), the current has the same direction as the electric field. As a result, white arrows in the ring could represent directions of the currents. In channel A, when the unit cell is illuminated by *X-*direction-linearly polarized (XLP) light, surface currents are almost localized in two legs of the equilateral triangular ring and flow from positive (+) to negative (−). There are pathways 1 and 2 from “+” to “−”. When the *X-*direction-linearly polarized light turns counterclockwise to *Y*-direction-linearly polarized (YLP) light by 90 degrees, currents exist in two legs and the base of the triangular ring. The positions of “+” and “−” turn counterclockwise by 90 degrees, and there are still two pathways 1 and 2 rotated by 90 degrees following the change in polarized light. In channel A, currents flow from “+” to “−” in two pathways, and the two pathways turn by 90 degrees with the change in the incident light from the *X-*direction to *Y*-direction. The two pathways are varied with the change in incident light, while the total amount of current inside the ring is almost unchanged. This guarantees a relatively stable reflectance and insensitivity to polarization angles in channel A. As for channel B, with *X-*direction-linearly polarized light, few currents exist in the triangular ring. The direction of the electric field is almost unchanged inside the triangular ring. However, with *Y*-direction-linearly polarized light, currents emerge and are mainly sustained in the base of the ring. Currents flow counterclockwise in the ring. In channel B, with the incident light rotated from the *X-*direction to *Y*-direction, currents inside the ring emerge. This leads to an increase in the reflectance and sensitivity to polarization angles in channel B.

### 3.2. Two-Bit Programmable Imaging

In Figure 4a, the reflectance spectra of the metasurface can be tuned with the change in the polarization angle. In Figure 6a, the metasurface is proposed for digital imaging according to Figure 4a. This metasurface is an 8 × 8 array of the unit cells with *K* = 2 μm. Quadrants 2 and 4 are formed by rotating quadrants 1 and 3 counterclockwise by 90 degrees. Figure 6b shows the magnetic field distributions when the metasurface is illuminated by *X-* and *Y*-polarized incident light. In channel A, it is shown that strong magnetic field distributions exist in all quadrants. As for channel B, strong magnetic distributions exist in quadrants 1 and 3 with *X-*direction-linearly polarized light, while the strong magnetic distributions exist in quadrants 2 and 4 with *Y*-direction-linearly polarized light. The quadrants with weak and strong magnetic field distributions can be coded as “0” and “1”, respectively. In Figure 6b, channel A shows a stable pattern as “1111” and channel B expresses the digital images as “1010” and “0101”.

To quantitatively assess the quality of the generated images, imaging efficiency η is introduced. $P_{i}$ represents the reflected power of each quadrant and $P_{r}$ denotes the reflected power of the metasurface. η is expressed as:(12)$$\eta =\frac{P_{i}}{P_{r}}$$

The imaging efficiency in Figure 6b can be calculated. In channel A, four quadrants all have η = 25% with both *X-* and *Y*-polarization, while in channel B, η of zone “0” and “1” are 14% and 36%, respectively.

### 3.3. Thermal Imaging

According to Figure 4c, the reflectance spectra of the metasurface are varied with change in the crystalline fraction of GST in the gap. As the crystalline fraction of GST is in direct proportion to the external temperature, we design a metasurface that can be used for thermal imaging. The structure has an array of 10 × 10 unit cells with *K* = 3 μm, as shown in Figure 7a. In Figure 7b, the crystalline fraction in each unit cell of the metasurface is shown. The crystalline fraction changes from 0 to 1 in a step of 0.25. The crystalline fraction distribution in Figure 7b is used to simulate thermal distribution on the metasurface.

In Figure 7b, the thermal imaging metasurface is divided into five parts according to different crystalline fractions. Reflectance spectra of five parts are shown in Figure 8a. For example, the black dotted line in Figure 8a denotes the reflectance spectrum of the part in which the crystalline fraction is 0.0 in Figure 7b. In channel A, the reflection peaks are almost localized at the wavelength λ_0_. However, in channel B, the reflection peaks increase from λ_1_ to λ_5_ with the change in crystalline fraction. In Figure 8b, for channel A, the magnetic field distribution at reflection peak λ_0_ is shown and the imaging efficiencies of crystalline fractions 0.00, 0.25, 0.50, 0.75, and 1.00 are 23%, 23%, 29%, 10%, and 15%, respectively. In Figure 8c–g, for channel B, strong magnetic distributions at reflection peaks λ_1_, λ_2_, λ_3_, λ_4_, and λ_5_ are mainly localized in the field of crystalline fractions of 0.00, 0.25, 0.50, 0.75, and 1.00, respectively. The maximal imaging efficiencies at reflection peaks λ_1_, λ_2_, λ_3_, λ_4_, and λ_5_ are 40%, 37%, 40%, 37%, and 46%, respectively. Each reflection peak could represent a crystalline fraction and a temperature. In this way, thermal imaging is realized because the thermal distribution can be reflected with magnetic field distributions at different reflection peaks in channel B. In other words, the thermal distribution on the metasurface can be illustrated by the reflected light distributions. 

In Table 1, we collect the previous works and compare the results with our work to show the advantages of the proposed structure in terms of wavelength range, functions, and imaging applications. Our work can be manipulated with dual channels, polarization angle, and temperature. In addition, two-bit coding and thermal imaging metasurfaces are achieved based on multiple manipulations. In other words, our designed metasurface is more tunable and can be designed for thermal imaging compared with others.

## 4. Conclusions

In conclusion, a dual-channel mid-infrared toroidal metasurface is designed with split equilateral triangular ring resonators. Calculated by TCMT, the toroidal dipole metasurface has two channels of which one is sensitive and another is insensitive to the polarization angle of incident light and temperature. A coding metasurface is realized with *X-* or *Y*-polarization expressed as “0” or “1” in digital mode. Another metasurface is proposed to achieve thermal imaging that can represent the external thermal distribution with a reflected wavelength distribution. The proposed metasurfaces offer alternative approaches to many information processing and energy-harvesting applications.

## Figures and Tables

**Figure 1 nanomaterials-12-03300-f001:**
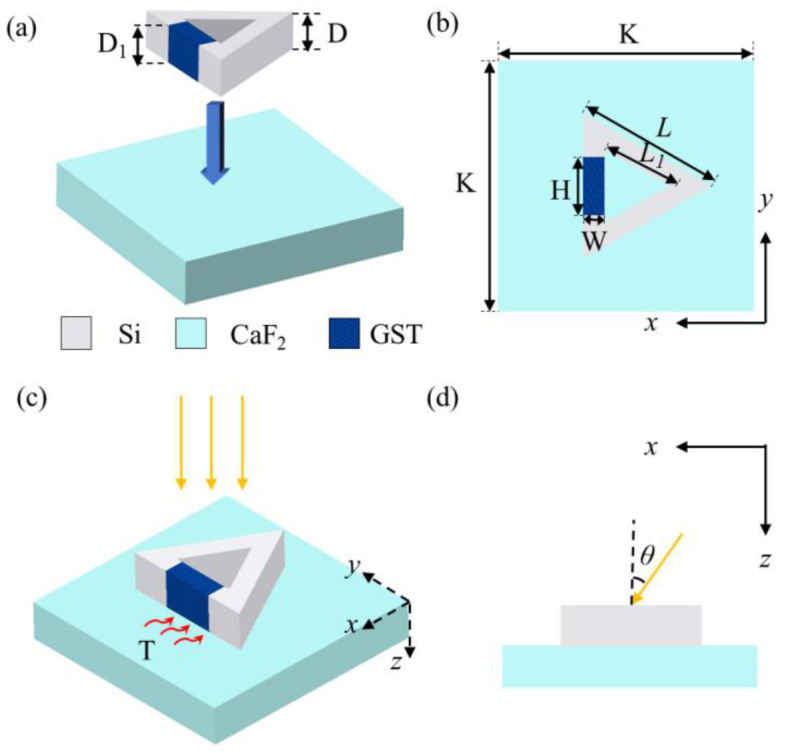
(**a**) Three-dimensional configuration of the unit cell; (**b**) top view of the unit cell; (**c**) configuration of the enlightened unit cell baked by an external heat source; yellow arrows indicate the linearly polarized incident plane wave; (**d**) incidence angle of the linearly polarized incident plane wave.

**Figure 2 nanomaterials-12-03300-f002:**
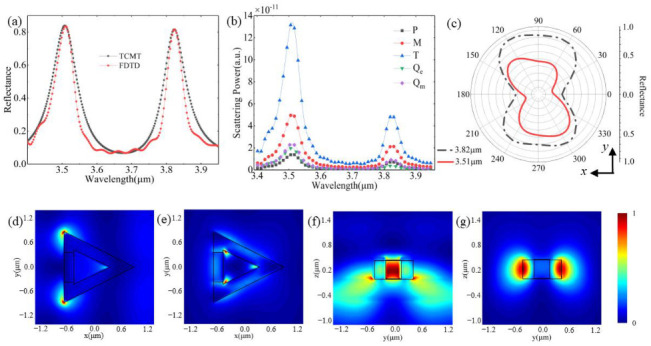
(**a**) Reflectance spectra of the metasurface obtained by FDTD and TCMT; (**b**) scattering powers of multipole radiating moments; (**c**) far-field distributions in *X-Y* plane at the wavelengths of 3.51 μm and 3.82 μm; normalized electric field intensity distributions in *X-Y* plane at the wavelengths of (**d**) 3.51 μm and (**e**) 3.82 μm; normalized electric field intensity distributions in *Y-Z* plane at the wavelengths of (**f**) 3.51 μm and (**g**) 3.82 μm.

**Figure 3 nanomaterials-12-03300-f003:**
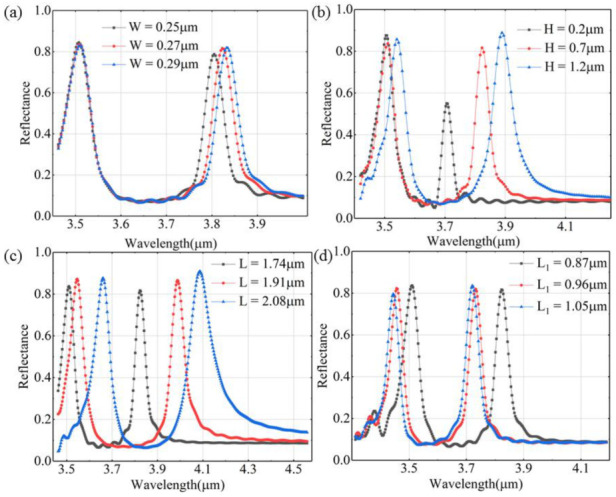
Reflectance spectra of different unit-cell geometrical parameters: (**a**) *W*; (**b**) *H*; (**c**) *L*; (**d**) *L*_1_.

**Figure 4 nanomaterials-12-03300-f004:**
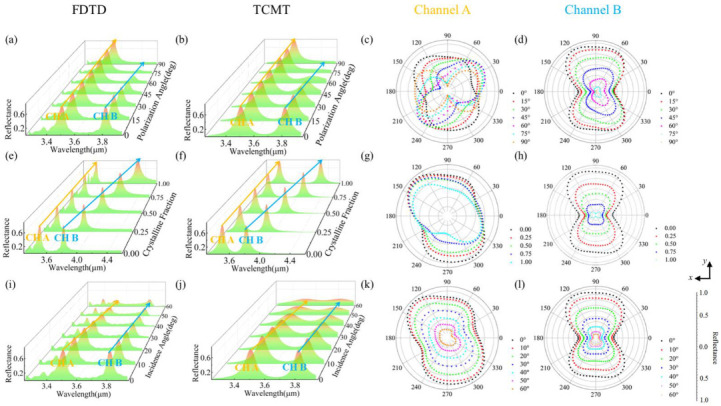
(**a**) Reflectance spectra with different polarization angles of incident light; (**b**) reflectance spectra with different polarization angles of incident light calculated by TCMT; (**c**) far-field scattering patterns in *X-Y* plane of channel A with different polarization angles; (**d**) far-field scattering patterns in *X-Y* plane of channel B with different polarization angles; (**e**) reflectance spectra with different crystalline fractions of GST; (**f**) reflectance spectra with different crystalline fractions of GST calculated by TCMT; (**g**) far-field scattering patterns of channel A in *X-Y* plane with different crystalline fractions; (**h**) far-field scattering patterns of channel B in *X-Y* plane with different crystalline fractions; (**i**) reflectance spectra with different incidence angle *θ*; (**j**) reflectance spectra with different incidence angle *θ* calculated by TCMT; (**k**) far-field scattering patterns in *X-Y* plane of channel A with different incidence angle *θ*; (**l**) far-field scattering patterns in *X-Y* plane of channel B with different incidence angle *θ*.

**Figure 5 nanomaterials-12-03300-f005:**
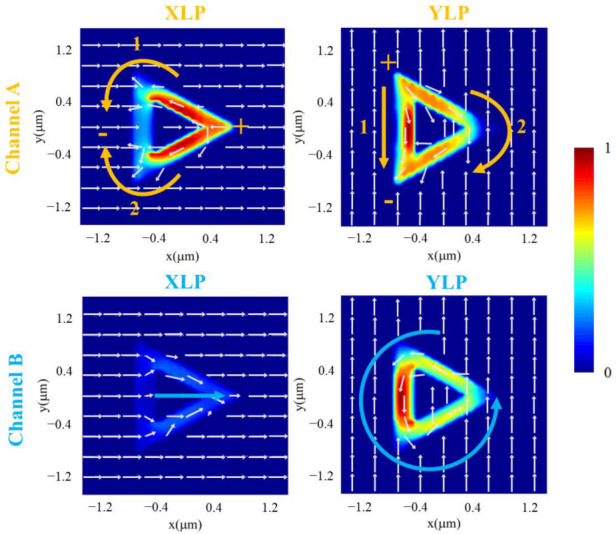
Surface current distributions with different channels and polarization angles. White arrows represent the directions of the electric field of the unit cell. Yellow arrows represent the directions of the currents in channel A. Blue arrows represent the directions of the currents in channel B.

**Figure 6 nanomaterials-12-03300-f006:**
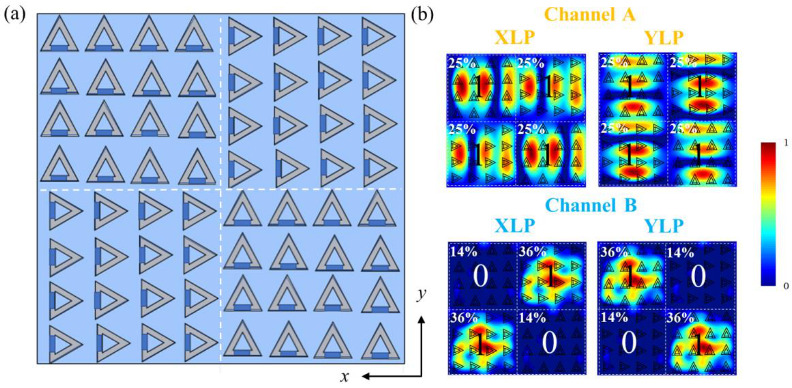
(**a**) Scheme of programmable metasurface; (**b**) magnetic field distributions of two channels under incident light with *X-*polarization and *Y*-polarization. The percentages represent the imaging efficiencies of the generated images.

**Figure 7 nanomaterials-12-03300-f007:**
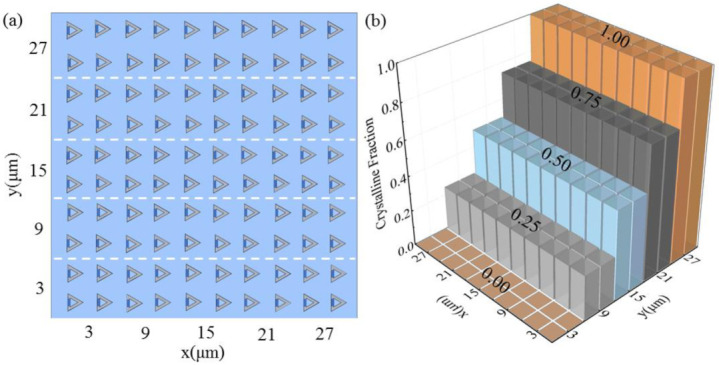
(**a**) Scheme of metasurface for thermal imaging; (**b**) crystalline fraction distribution on the metasurface.

**Figure 8 nanomaterials-12-03300-f008:**
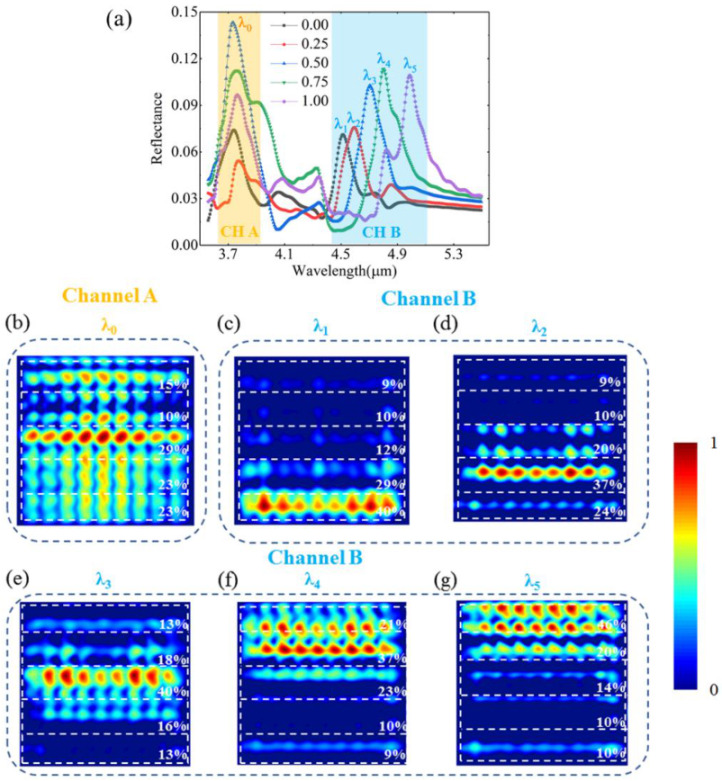
(**a**) Reflectance spectra of different crystalline fraction areas; (**b**) magnetic field distribution in channel A at λ_0_; magnetic field distributions in channel B at (**c**) λ_1_, (**d**) λ_2_, (**e**) λ_3_, (**f**) λ_4_, and (**g**) λ_5_.

**Table 1 nanomaterials-12-03300-t001:** Comparison of our work with analogous structures.

Ref.	Structure	Range	Tunability	Imaging Application
[8]	graphene–metal split ring resonators	0.3–0.6 mm	① circular dichroism ② graphene	2-bit switchable coding
[9]	graphene–metal open rings	0.29–0.32 mm	① circular dichroism ② graphene	① 1-bit coding ② 2-bit coding
[10]	circular ring with parabolic-hole	1–7 μm	① polarization angle ② wavelength	① 1-bit coding ② 2-bit coding
[11]	two circular rings and crossed slots	9–20 mm	① polarization angle ② wavelength	smile-face image
Ours	split equilateral triangular rings	3–4 μm	① dual channel ② polarization angle ③ GST	① 2-bit coding ② thermal imaging

## Data Availability

The available data have already been stated in the article.

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
