# Peer review of "Dual-Channel Mid-Infrared Toroidal Metasurfaces for Wavefront Modulation and Imaging Applications"

_nanomaterials, 2022, doi:10.3390/nano12193300_

Round 1

Reviewer 1 Report

Please find attached a pdf file with my comments.

Best Regards.

Reviewer 2 Report

The authors study a metasurface able to work at two wavelengths in the mid-IR for wavefront modulation and imaging. The subject of the paper is very interesting to the scientific community, however the presentation of the results must be improved.

- For the three potential applications, i.e. wavefront modulation, 2-bit programming imaging and thermal imaging, there is no simple example able to show the real potential of the proposed metasurface. I suggest to add, for any application, a simple proof-of-principle example to show numerically the functioning, e.g. beam steering for wavefront modulation and a simple imaging example.

- In the introduction section, the pertinent state of the Art must be expanded to include all papers studying similar devices or applications, an the different approaches proposed. The table for comparison can be left in the conclusions.

- How initial geometrical parameters have been chosen?

- For the sake of the completeness, the authors must discuss on a possible path for fabrication and realization of the proposed device.

Round 2

Reviewer 1 Report

I evaluated the new version of the manuscript “Dual-channel mid-infrared toroidal metasurfaces for wavefront modulation and imaging applications” by J. Zhang et al.

The manuscript has considerably improved and deserves publication. Indeed, it may be of interest for the readers of MDPI Nanomaterials.

Just a minor suggestion:

Lines 35-37 “The tunability can be realized by introducing external excitation methods are used including electric [20], optical [21-23] and mechanical controlling [24], etc.”. Maybe the authors could also mention the thermal tuning, i.e. the change of the optical response by varying the system temperature. As a starting point I put a couple of references that may be added on this topic:

https://doi.org/10.1002/adfm.201700580

https://doi.org/10.1364/OE.440564

Thank you very much for your attention. Best Regards.
